# Molecular Insights into the Mechanism of Necroptosis: The Necrosome as a Potential Therapeutic Target

**DOI:** 10.3390/cells8121486

**Published:** 2019-11-21

**Authors:** Jing Chen, Renate Kos, Johan Garssen, Frank Redegeld

**Affiliations:** 1Division of Pharmacology, Utrecht Institute for Pharmaceutical Sciences, Faculty of Science, Utrecht University, 3508 TB Utrecht, The Netherlands; j.chen@uu.nl (J.C.); r.kos@amsterdamumc.nl (R.K.); j.garssen@uu.nl (J.G.); 2Danone Nutricia Research, Uppsalaan 12, 3584 CT Utrecht, The Netherlands

**Keywords:** necroptosis, MLKL, RIP kinase, necrosome, programmed cell death, cancer, inflammatory diseases

## Abstract

Necroptosis, or regulated necrosis, is an important type of programmed cell death in addition to apoptosis. Necroptosis induction leads to cell membrane disruption, inflammation and vascularization. It plays important roles in various pathological processes, including neurodegeneration, inflammatory diseases, multiple cancers, and kidney injury. The molecular regulation of necroptotic pathway has been intensively studied in recent years. Necroptosis can be triggered by multiple stimuli and this pathway is regulated through activation of receptor-interacting protein kinase 1 (RIPK1), RIPK3 and pseudokinase mixed lineage kinase domain-like (MLKL). A better understanding of the mechanism of regulation of necroptosis will further aid to the development of novel drugs for necroptosis-associated human diseases. In this review, we focus on new insights in the regulatory machinery of necroptosis. We further discuss the role of necroptosis in different pathologies, its potential as a therapeutic target and the current status of clinical development of drugs interfering in the necroptotic pathway.

## 1. Introduction

Programmed cell death can be regulated by different intrinsic cell death programs. It plays important roles in multiple physiological processes, including organ development, homeostasis and resilience of the immune system, and disease pathogenesis. Initially, apoptosis was defined as the only form of programmed cell death, whereas necrosis was regarded as unregulated and resulted from a damaging environmental stress response. Not until 2005, it was found that there is a regulated form of necrosis termed necroptosis, which can also be controlled by specific intrinsic programs [1]. Necroptosis can be described as cellular-suicide because of its role in several viral infections where it initiates cell death to diminish virus replication [2]. Since necroptosis has been discovered shortly, there are still many questions to be clarified on the molecular mechanism and key regulatory factors.

Necroptotic cells undergo very unique morphology changes which are distinctive from apoptosis as listed in Table 1 [3]. During necroptosis, cells endure a rapid swelling, with simultaneous swelling of organelles, rather than the cell shrinkage which occurs during apoptosis. Also, instead of blebbing of the plasma membrane, necroptotic cells undergo early plasma membrane disruption, phosphatidylserine exposure and organelle breakdown, leading to the leakage of intracellular contents and as a consequence triggering inflammation. Unlike necroptosis, necrosis is an unregulated form of cell death that occurs as a consequence of damaging environmental stress, such as extreme physiological stress and viral or toxin-mediated infections. Morphological features of necrosis are similar to necroptosis, including cell and organelle swelling that lead to release of intracellular contents and pro-inflammatory response.

Necroptosis can be induced by various stimuli, such as tumor necrosis factor α (TNFα), interferon γ, lipopolysaccharide (LPS), pathogen- and damage-associated molecular patterns [4]. Noteworthily, necroptosis and apoptosis can be induced by almost the same range of stimulating factors, indicating the importance of additional regulators to initiate the specific necroptotic pathway. The necroptotic pathway is regulated by multiple steps of post-transcriptional modifications including phosphorylation and ubiquitination [5,6]. The core of the necroptosis mechanism is regulation of the formation of necrosome, a complex consisting of RIPK1, RIPK3 and MLKL [7].

Besides its role of inducing cell death, necroptosis can provoke an adaptive immune response by driving cytokine secretion asserting pro-inflammatory effects as well [8]. Necroptosis is implicated in the development of pathologies characterized by unwarranted cell loss with an inflammatory component [9]. Moreover, necroptosis is deemed to be protective against certain malignancies and to improve disease outcome [10,11].

Better understanding of the role and regulatory mechanisms of necroptosis in disease development and progression would help to advance novel mechanism-based therapeutics. In this review, we will discuss the recent findings concerning necroptosis and necrosome components. Here, we mainly focus on the activation and regulation mechanisms of key proteins which are involved in necroptosis. Furthermore, we will discuss the potential therapeutic opportunities to intervene in or to stimulate necroptosis.

## 2. Necroptosis Induction

Although the necroptotic pathway can be initiated by different stimuli, the molecular mechanism of tumor necrosis factor receptor 1 (TNFR1)-mediated necroptosis remains the best-understood and therefore this pathway will be further discussed to illustrate the key signalling components.

### 2.1. Pro-Survival Signalling Complex I

TNFα is an important pro-inflammatory cytokine which is known to be associated with multiple inflammatory diseases including cancer. Upon binding of TNFα to TNFR1, the receptor has been found to trigger a rapid formation of a high molecular weight complex, named complex I, by recruitment of multiple proteins after which TNFR1 undergoes a conformational change [12]. These recruited proteins include: RIPK1, TNFR-associated death domain (TRADD), TNFR-associated factor 2 (TRAF2), TRAF5, and the E3 ubiquitin ligases cellular inhibitor of apoptosis 1 (cIAP1), cIAP2 and linear ubiquitin chain assembly complex (LUBAC). TRADD is recruited to TNFR1 through their death domains (DD) and the initial signal transduction [13]. TRAF2/5 and RIPK1 are subsequently recruited, which was found to be mediated via direct interaction with TRADD. The TRAF proteins then recruit E3 ligases cIAP1/2 and LUBAC to RIPK1. TRAF2 was found to stabilize cIAP1 by inhibiting its autoubiquitination and proteasomal degradation, thus facilitating the recruitment of cIAP1 [14]. Both cIAP1/2 and LUBAC catalyse the Lys63-linked polyubiquitination of the intermediate domain of RIPK1. NEMO (NF-κB essential modulator), also known as IKKγ, then binds specifically to the Lys63 polyubiquitin chains of RIPK1 via a specific ubiquitin-binding domain, leading to subsequent recruitment of the IĸB kinase (IKK) complex (IKKα and IKKβ) [15]. The polyubiquitination of RIPK1 is also essential for recruitment of the transforming growth factor β-activated kinase (TAK) complex [TAK1 and TAK1 binding protein (TAB) 1/2] [16]. Both the TAK and the IKK complexes activate the nuclear transcription factor-kappa B (NF-κB) pathway which promotes cell survival by activating the expression of multiple pro-inflammatory and pro-survival genes, such as cellular FLICE-like inhibitory protein (cFLIP). Furthermore, phosphorylation of RIPK1 at Ser25 by the IKKα/β directly inhibits RIPK1 kinase activity and prevents the assembly of the death-inducing signalling complex, also known as complex II [17,18]. Thus, the formation of complex I promotes cell survival through activating the NF-κB pathway, as schematically depicted in Figure 1.

### 2.2. Cell Death-Inducing Signalling Complex II

The transition of complex I to complex II is regulated through the RIPK1 polyubiquitination profile. Deubiquitinating enzymes cylindromatosis (CYLD) and A20, both hydrolyse Lys63-linked ubiquitin chains, are associated with the regulation and formation of the complex II [19]. Complex II abides in two different forms (IIa and IIb) depending on the protein composition and activity. It has been found that both complexes are capable of inducing apoptosis or necroptosis depending on the cell environment, as schematically depicted in Figure 1 [20,21]. Complex IIa consists of TRADD, RIPK1, FAS-associated protein with a death domain (FADD) and caspase-8. Within hours after TNFα stimulation, complex IIa has been found to be formed after recruiting deubiquitinases, such as CYLD and A20, leading to the release of TRADD and RIPK1 from TNFR1 [19,21]. Subsequently, FADD and pro-caspase-8, both proteins associated with cell death, are recruited to complex IIa, resulting in activation of caspase-8 by cleavage [12]. The activated caspase-8-induced apoptosis is independent of RIPK1 kinase activity in complex IIa [21]. On the other hand, the absence of cIAPs (IAP antagonist or knockdown of cIAPs), and inhibition of TAK1 or IKK complex allow for the formation of complex IIb, which does not contain TRADD but still contains RIPK1, FADD, and caspase-8 [22]. In the absence of TRADD, RIPK1 kinase activity is required for caspase-8 activation to induce RIPK1-dependent apoptosis. In both complex IIa and IIb, when the levels of RIPK3 and MLKL are sufficiently high and caspase-8 is inactivated or absent, necroptosis is induced by recruitment of RIPK3 followed by necrosome formation, which is the core of the necroptosis machinery consisting of RIPK1, RIPK3 and MLKL [20,21].

Hence, TNFR1 complex I triggers NF-κB activation and promotes cell survival. Complex IIa leads to caspase-8-mediated (RIPK1-independent) apoptosis. Complex IIb results in caspase-8-mediated apoptosis, which depends on RIPK1 kinase activity. When caspase-8 is absent or inhibited, the necrosome is formed and leads to MLKL-mediated necroptosis.

#### Inhibition of Caspase-8 Activity Is Essential for Necroptosis

Caspase-8 is known to initiate apoptosis downstream of death receptors such as CD95/FasR, TNFR1 and DR4/DR5 by undergoing autocatalytic activation and activating the executioner caspases. Initially, caspase-8 is recruited to FADD in its pro-caspase-8 form, leading to recruitment of more pro-caspase-8 molecules to undergo homodimerization [23]. Then, apoptosis is initiated by the cleavage of two pro-caspase-8 proteins at aspartates resulting in the maturation of pro-caspase-8 into caspase-8 [24]. Subsequently, caspase-8 activates caspase-3/7 leading to apoptosis.

The association of the anti-apoptotic protein cFLIP to the FADD-pro-caspase-8 complex has been found to determine cell fate, as depicted in Figure 1 [25]. There are two isoforms of cFLIP proteins, the long (cFLIP_L_) and short (cFLIP_S_) forms, and they exist different functions in the determination of cell fate. cFLIP_S_ functions as a natural inhibitor of caspase-8 by blocking the oligomerization of pro-caspase-8 [24]. cFLIP_L_ was reported to enhance apoptosis if present at a low level. However, if it is present in a high level, cFLIP_L_ could act as an inhibitor of caspase-8 [25]. Nonetheless, the FADD-pro-caspase-8-cFLIP_L_ complex inhibits activation of RIPK3, which subsequently inhibits necroptosis [26]. Furthermore, inhibiting caspase-8 activity, for example by pan-caspase inhibitor zVAD-fmk or viral cFLIP_S_ mimetics, is essential for initiating RIPK1-mediated necroptosis, as schematically depicted in Figure 1.

## 3. RIPK1 Has Multiple Functions

Structurally, RIPK1 consists of an N-terminal kinase domain, an intermediate domain containing a RIP homotypic interaction motif (RHIM), and a C-terminal death domain with 6 α-helices [27,28].

RIPK1 can initiate cell-death through three pathways, which is regulated by its phosphorylation level by TAK1 (Figure 2) [29]. Transient phosphorylation of RIPK1 by TAK1 at Ser321 in the intermediate domain upon TNFα stimulation activates the RIPK1-independent apoptosis, which requires TRADD and formation of complex IIa. In the absence of TAK1 (TAK1 knockout, inhibitor of TAK1 or IAP antagonist), the intermediate domain cannot be phosphorylated. The unphosphorylated RIPK1 triggers RIPK1-dependent apoptosis via recruitment of FADD and formation of complex IIb, which is independent of TRADD. Moreover, in TNFα-treated A20/TAB2-deficient cells, hyperphosphorylation of RIPK1 by unregulated TAK1 recruits RIPK3 and induces MLKL-mediated necroptosis [29].

Furthermore, RIPK1 can also be activated to induce necroptosis via auto-phosphorylation by its kinase function itself [30]. Upon blockage of the RIPK1 kinase activity via an Asp138Asn mutation in a mouse model (Ripk1D138N/D138N mice), fibroblasts are protected against necroptosis but induction of similar levels of apoptosis like in wildtype was observed. The kinase-inactive form of RIPK1 was found to protect against necroptotic cell death [31,32]. All these findings suggest that the kinase activity of RIPK1 is essential for activating the necroptotic pathway.

Moreover, the dimerization of the DD is essential for necroptosis induction. This dimerization is dependent of Lys599 in human (Lys843 in murine species), and Lys599Arg/Lys843Arg mutations blocked induction of necroptosis both in vitro and in vivo [33].

Finally, ubiquitination of RIPK1 is essential for its regulation and the interaction with RIPK3. Besides the polyubiquitination by cIAPs and LUBAC to form complex I, and the deubiquitination by CYLD and A20 to form complex II, pellino E3 Ubiquitin Protein Ligase 1 (PELI1) also regulates the function of RIPK1 [34]. Deficiency of PELI1 disrupts the interaction between RIPK1 and RIPK3 and abrogates necroptosis, suggesting that PELI1 might function as a mediator in this process. PELI1-induced Lys63 ubiquitination of RIPK1 on Lys115 was confirmed, which was dependent of the RIPK1 kinase activity [34]. Taken together, this implies that phosphorylation, dimerization and ubiquitination of RIPK1 are essential for inducing necroptosis.

In addition to its kinase activity, RIPK1 was reported to show other miscellaneous functions [35]. Several studies have been attempted to elucidate triggers and functionalities of RIPK1 [36,37]. It has been shown that knockdown of RIPK1 cannot prevent the LPS-induced necroptosis, but did inhibit TNFα-induced necroptosis, whereas RIPK3 knockdown prevented both LPS- and TNFα-induced necroptosis [36]. These findings suggest that activation of necroptosis is not fully dependent on RIPK1. Besides the known role of RIPK1 in activating necroptosis, RIPK1 can also suppress the necroptosis by inhibiting other RIPK3 potentiators [37]. This is regulated via the RHIM domain of RIPK1, which mediates the protein interactions with RIPK3, as it competes and interacts with other RHIM domain containing proteins. Z-DNA binding protein-1 (ZBP1) is a RHIM domain containing protein that can be induced by IFNs. It was found to be involved in viral DNA-mediated type I IFN induction and NF-κB activation. Upon mutation of the RHIM domain of RIPK1, ZBP1 binds to RIPK3 and induces necroptosis; reconstitution of the RIPK1 RHIM domain is protective against this and blocks subsequent cell death [37,38]. This suggests that the RHIM domain of RIPK1 shows a protective role against ZBP1-mediated necroptosis. Another function of RIPK1 is exerted independent of its kinase activity. If the kinase-inactivated RIPK1 (with Asp138Asn mutation) is present, mice with such mutation are still viable rather than mice which are RIPK1 deficient [31]. Mice lacking RIPK1 kinase activity were also shown to be resistant to the induction of immune-mediated liver injury similar to wild type mice. In contrast, RIPK1-deficient mice were sensitive to the immune-mediated liver injury [39]. Thus, it suggests that RIPK1 might have a kinase-independent function that is able to protect to injury. In summary, RIPK1 can induce apoptosis or necroptosis, inhibit necroptosis, and exerts kinase-independent protective functions under specific conditions.

Currently, necrostatin-1 (Nec-1) is the most potent and specific inhibitor of RIPK1 and it is commonly used in necroptosis inhibition studies (Table 2) [1]. RIPA-56, a novel small molecule, has been developed to inhibit RIPK1 by direct targeting resulted in inactivating the kinase activity [40]. RIPK1 inhibitors such as GSK2982772 and GSK3145095 are currently evaluated in several clinical trials in rheumatoid arthritis (NCT02858492), psoriasis (NCT02776033), ulcerative colitis (NCT02903966) and cancer (NCT03681951) (Clinicaltrials.gov). Another RIPK1 inhibitor DNL747 is currently in clinical development for the treatment of neurodegenerative diseases including Amyotrophic lateral sclerosis (NCT03757351) and Alzheimer’s disease (NCT03757325) (Table 3).

## 4. RIPK3 Plays a Key Role in Necroptosis

Like RIPK1, RIPK3 has an N-terminal kinase domain, followed by an intermediate domain containing the RHIM motif. Unlike RIPK1, RIPK3 does not have a C-terminal death domain [27]. The RHIM motif, which is also present in the intermediate domain of RIPK1, mediates the interaction with RIPK1 and then induces necrosome formation.

Upon necroptosis induction, RIPK3 is recruited to RIPK1 and then becomes phosphorylated and activated, leading to necrosome formation and eventual to cell death. The formation of the initial heterodimer of RIPK1 and RIPK3 recruits more RIPK3 and induces homodimerization of RIPK3, which leads to its auto-phosphorylation and activation. Phosphorylation of RIPK3 at Ser227 (Thr231/Ser232 for mouse RIPK3) plays a crucial role in recruiting and activating its downstream substrate MLKL. The mechanism by which RIPK3 induces cell death is depicted in Figure 3. The activity of RIPK3 is also controlled by its phosphorylation. Recently, a disintegrin and metalloprotease 17 (ADAM17) has been shown to be essential in this phosphorylation process. The phosphorylated-RIPK3 then recruits and phosphorylates MLKL, eventually leading to necroptosis. Knockdown of ADAM17 could block RIPK3 phosphorylation and subsequent necroptosis [51]. Nonetheless, upon blockage of the kinase activity of RIPK3 via an Asp161Asn mutation in a mouse model, mice are unable to survive due to induction of RIPK1-dependent apoptosis [32]. This indicates that another kinase-independent trigger of RIPK3 induced cell death, as viral infection lead to FADD-dependent, but RIPK3-independent apoptosis [44].

Furthermore, ubiquitination also regulates RIPK3 activity in necroptosis. RIPK3 is ubiquitinated at Lys5 after necroptosis induction. RIPK3 with Lys5 mutated to Ala is not able to be ubiquitinated and form RIPK1-RIPK3 complex, resulting in the inhibition of necroptosis [52]. The ubiquitination of RIPK3 can be abrogated by A20 through its deubiquitinating motif. Higher level ubiquitination of RIPK3 and faster assembly of RIPK1-RIPK3 heterodimer were observed in A20-deficient cells, indicating that A20 also plays a negative role in necroptosis besides its role in the complex II formation [52]. In addition, RIPK3 can be ubiquitinated at Lys55 and Lys363 by chaperone-assisted E3 ligase C-terminus of Hsp70-interacting protein (CHIP). This ubiquitination functions as a lysosomal degradation signal for RIPK3 and prevents necroptosis. Knockdown of CHIP enhanced TNFα-induced necroptosis [53]. This finding suggests that CHIP acts as a negative regulator in necroptosis.

Moreover, proteasome inhibitors MG132 and bortezomib were found to induce RIPK3-dependent necroptosis in mouse fibroblasts and human leukemia cells, which was independent of kinase activity but acted via the RHIM domain [54]. Proteasome inhibition induces the accumulation of Lys48-linked polyubiquitination of RIPK3 at Lys264, perhaps, resulting in release of the RHIM domain. Then released-RIPK3 forms a homo-oligomer and further recruits MLKL to trigger necroptosis. Unlike TNFR-mediated necroptosis, proteasome inhibitor-induced necroptosis does not require caspase-8 inhibition [54]. Hence, depending on the position of the ubiquitination, it can either tag RIPK3 for degradation or prime RIPK3 for necroptosis initiation.

Hence, phosphorylation and ubiquitination play pivotal roles in RIPK3-mediated necroptosis. After activation of RIPK3, in the absence of caspase-8 activity, MLKL is recruited to RIPK3 and induces necroptosis [55]. For necroptotic induction, caspase-8 activity must be blocked (the exception being with influenza A virus as referenced by Nogusa et al.) and that would also block apoptosis.

GlaxoSmithKline has developed a class of potent RIPK3 inhibitors to rescue cells from necroptotic cell death, including GSK’840, GSK’843 and GSK’872 [44,45]. Furthermore, the B-Raf inhibitor dabrafenib has been found to be able to inhibit RIPK3 activity and rescue necroptosis but not apoptosis [46]. GW440139B, screened from a library of 8904 bioactive compounds, was identified as a RIPK3 inhibitor by blocking RIPK3-mediated MLKL phosphorylation and inhibiting the following MLKL oligomerization [56]. Dabrafenib has been advanced into clinical trials for melanoma treatment (Table 3). Recently, HS-1371 was identified as a novel specific RIPK3 inhibitor which binds to the ATP-binding pocket of RIPK3 and inhibits kinase activity [47]. In addition, ponatinib and pazopanib have been found to specifically block necroptosis, but not apoptosis by targeting RIPK1/3 in human cells [43], and they were also recently proposed as possible candidates in clinical therapy of various cancer types as listed in Table 3.

## 5. MLKL Is the Effector Protein in Necroptosis

Structurally, MLKL consists of an N-terminal four-helical bundle domain (NTD), a brace region (BR) with two α-helices, and a C-terminal kinase like domain (KLD) containing a helical activation loop and an ATP binding pocket [57].

MLKL functions as an executor of necroptosis by its oligomerization and membrane translocation, and subsequent formation of membrane-disrupting pores (Figure 4). Intriguingly, recent work showed that the NTD of MLKL is required and sufficient to induce its oligomerization and trigger necroptosis by ectopic expression of truncated MLKL in human cells [58]. Furthermore, this mechanism was also demonstrated in murine cells [59]. However, so far, there is no information about whether the N-terminal fragment of MLKL is formed and present endogenously to induce necroptosis under (patho)physiological conditions.

The phosphorylation, oligomerization and membrane translocation of MLKL are key events to execute necroptosis. Phosphorylated MLKL is sufficient to activate necroptosis when RIPK3, MLKL and other necroptotic stimuli are not present [60]. The murine MLKL can be phosphorylated at Ser345, Ser347 and Thr349 in the KLD by RIPK3; and for human MLKL at Thr357 and Ser358. Single-site mutation analysis in murine cells showed that phosphorylation of Ser345 is critical for the RIPK3-mediated MLKL activation and the following necroptosis, whereas phosphorylation of Ser347 and Thr349 are dispensable for its activation. Nonetheless, the phosphorylation at Ser347 of MLKL did aid in activation of necroptosis [44]. Upon phosphorylation, MLKL undergoes conformational changes resulting in the unfolding and release of the BR. MLKL finally forms oligomers and incites binding of the NTD to the plasma membrane [61]. The phosphorylation-dependent oligomerization of MLKL starts from forming tetramers which serves as building blocks for larger polymers [62]. It has been reported that MLKL forms an octamer through two tetramers with disulphide bonds at α-helices 4 and 5 within the KLD [63]. These disulphide bonds are only required for MLKL tetramerization but not octamerization which is essential for necroptosis. These octamers are formed in the necrosome in association with RIPK3 and are thereafter released to translocate to the membrane [63]. At present our understanding of the mechanism of MLKL oligomers induced-membrane disruption is very limited and remains unclear. Some studies showed that the NTD of MLKL binds to the plasma membrane directly through low-affinity phospholipid binding sites. The initial bound MLKL undergoes a conformational change and provides additional higher-affinity phospholipid binding sites. This induces a robust association of MLKL to the plasma membrane [61]. Calcium and sodium influx [64,65] were also reported to be required in MLKL mediated-plasma membrane rupture during necroptosis. The MLKL octamer spans across the plasma membrane with its N-terminal and C-terminal ends at the intracellular side [63]. So, after phosphorylation and oligomerization of MLKL, it can exert its pore-forming function to disrupt the plasma membrane.

Apart from the membrane translocation of MLKL, phosphorylated MLKL can translocate into the nucleus along with RIPK1 and RIPK3. This nuclear translocation is caused by the conformational changes after phosphorylation, which exposes the C-terminal nuclear localization sequence that allows MLKL to be imported into the nucleus [66]. The nuclear translocation of activated MLKL is not required for necroptosis, but contributes to necroptosis indirectly by regulating cytosolic necrosome formation and subsequent cell death [67].

Besides the phosphorylation-dependent activation, MLKL could potentially regulate necroptosis via its ATP-binding pocket within the KLD. The Glu351Lys mutation of human MLKL, which was discovered to lead to enhancement of ATP binding, disrupts the formation of MLKL tetramers and thus subsiding octamers. On the contrary, phosphomimic mutations at Thr357 and Ser358 in the human MLKL stabilize the tetramers, indicating that phosphorylation of MLKL protects from ATP-induced dissociation of oligomers [68]. However, human MLKL with mutations of Lys230Met/Gln356Ala in its ATP-binding pocket can still activate itself and translocate to the plasma membrane to induce necroptosis [69].

Recently, inositol-polyphosphate (IP) kinases was also found to be able to regulate necroptosis. Cells with depletion of IP multikinase (IPMK) and inositol-tetrakisphosphate 1-kinase (ITPK1) which are critical for synthesis of cellular IPs were unable to show MLKL membrane association by necroptosis induction, despite the presence of phosphorylated MLKL. In IPMK and ITPK1 complemented cells, the membrane disruption and subsequent necroptosis were restored [70]. Thus, this indicates that IP kinases regulate necroptosis by influencing the plasma membrane association of phosphorylated MLKL.

Besides the role of MLKL as executioner of necroptosis by disrupting the plasma membrane, it was also found to have other functions. MLKL-depleted cells showed a marked reduction in the rate of intracellular degradation of TNF after its binding to the TNF receptor in the absence of RIPK3, which led to alteration of NF-ĸB activity and the induction of TNF-induced genes including some inflammatory genes [71]. Also, MLKL depletion can slow down the intracellular degradation of epidermal growth factor receptor, which indicates a mechanism that MLKL might exert a general effect on receptor uptake and degradation. In addition, MLKL has been found to regulate endosomal trafficking, which is mediated independently of its phosphorylation by RIPK3. Furthermore, RIPK3 boosts release of phospho-MLKL-containing vesicles from cells. This release appears to withhold necroptotic cell death and might contribute to cellular communication [71]. Consequently, MLKL exerts a dual function both excitatory and inhibitory of cell-death, possibly indicating a feedback loop regulating cell fate.

Over the past years several MLKL inhibitors have been developed. The most commonly used MLKL inhibitor is necrosulfonamide (NSA), which binds to Cys86 of human MLKL and forms a covalent adduct to block MLKL oligomerization and inhibits subsequent necroptosis [49]. Recently, thioredoxin-1 (Trx1), a small redox protein, was reported as a MLKL inhibitor. Trx1 binds to monomeric MLKL molecules and prevents the oligomerization of monomeric MLKL and following necroptosis [72]. These inhibitors abrogate the plasma membrane association, which is vital for MLKL functioning. GW806742X was found to be able to directly bind the KLD of MLKL and therefore block MLKL phosphorylation, oligomerization and plasma membrane translocation [48].

In addition, heat shock protein 90 (HSP90), a molecular chaperone which functions in the stabilization and activation of multiple kinases, was reported to be essential for necroptosis by direct interacting with MLKL and stabilizing necroptotic proteins [73]. It was identified as an indispensable assistant for the necrosome assembly. HSP90 inhibitor 17AAG inhibited TNFα-induced necroptotic cell death by blocking the necrosome formation, MLKL oligomerization and subsequent membrane translocation [50]. Hence, interfering with HSP90 activity by specific inhibitors might be a new therapeutic approach for the treatment of necroptosis-associated human diseases. HSP90 inhibitors, including 17AAG and IPI-504, have been tested in clinical trials in patients with range of cancer types (Table 3).

## 6. The Necroptotic Pathway as Therapeutic Target

Necroptosis is a tightly regulated process and faulty regulation of necroptosis is implicated in the development of various human diseases associated with unwarranted cell loss and inflammatory responses [9]. In the next sections, we will summarize the activation and regulation of necrosome components in some necroptosis-associated diseases and discuss the latest insights into the therapeutic opportunities.

### 6.1. Necroptosis in Neurodegenerative Diseases

Necroptosis has been implicated in the development of some neurodegenerative diseases [74]. Alzheimer’s disease (AD) is a degenerative brain disease and is characterized by the damage and loss of neurons. It was shown that necroptosis was activated in human AD brain [41]; as well as in the brain of an AD mouse model [75]. The level of necroptosis markers including RIPK1, MLKL, necrosome complex and MLKL oligomers were significantly higher in AD brains [41]. Treating mice bearing AD brain with the necroptosis inhibitor 7-Cl-O-necrostatin remarkably suppressed necroptosis and prevented neuronal loss [41]. This suggests that interfering in necroptosis through inhibiting activities of necrosome components could be an interesting new strategy in the treatment of AD.

Amyotrophic lateral sclerosis (ALS) is a progressive neurodegenerative disease with the loss of motor neurons. Compared to healthy control spinal cords, the ALS spinal cords showed a significant increase in necrosome components including RIPK1, RIPK3 and MLKL in a mouse model of ALS [76]. Furthermore, loss of optineurin, an established ALS-related gene, caused sensitization for necroptosis. Inhibition of RIPK1 by Nec-1 or knock-out of RIPK3 protected against demyelination and lead to a decrease in axonal pathology hallmarks, in both a sporadic (optineurin-deficient) and familial (SOD1G93A) ALS mouse model [76]. This indicates that the necroptosis is activated in ALS and inhibition of necroptosis could be of potential therapeutic value. Moreover, upon inhibition of RIPK1 by Nec-1, neuronal degradation was reduced to only half compared to untreated cells in a Parkinson disease (PD) model [77]. The necroptosis marker p-MLKL has been observed to be activated in the brain of a PD mouse model and PD patients [78,79]. Thus, RIPK1 and RIPK3, by promoting inflammation and cell death, induce axonal degradation and act as mediators of ALS and PD. Moreover, the protective effect of Nec-1 indicates the therapeutic potential of this drug in ALS and PD. Likewise, inhibition of RIPK3 could possibly be exploited for treatment of these diseases.

Multiple Sclerosis (MS) is another chronic neurodegenerative disease, which is characterized by the loss of oligodendrocytes and demyelination. Necroptosis markers, including phosphorylation of RIPK1, RIPK3 and MLKL, were highly detected in pathological samples from MS patients, and there was a pronounced increase of MLKL oligomers in MS pathological samples compared to control [80]. This indicates that necroptosis is involved in the pathogenesis of MS. Oral administration of RIPK1 inhibitor 7-Cl-O-Nec-1inhibited oligodendrocyte degeneration and reduced the disease severity in a mouse model of MS [80]. In a recent study, it was shown that inhibition of RIPK1 by RIPA-56 inhibited progression of demyelination and disease development in a cuprizone-induced model for MS. Interestingly, a RIPK3-independent function of MLKL also appeared to be involved in the demyelination process [81]. These finding reveals that inhibiting RIPK1 might be a valuable therapeutic option in the treatment of MS.

In hernia disease, nucleus pulposus cells under mechanical stress by compression showed unique necroptosis hallmarks such as severe vacuolation and serious disruption of the plasma membrane. This indicates that apart from the inflammatory profile often described, necroptosis also plays a vital role in inducing cell death during mechanical stress. Specific inhibitors of the necrosome components, such as Nec-1 of RIPK1, GSK’872 of RIPK3 and NSA of MLKL, all showed protective effects on cell death under mechanical stress, with reduced necroptosis hallmarks and increased cell viability [82]. Therefore, inhibitors targeting various players in the necrosome might have a therapeutic value for the treatment of hernia diseases with cell death due to mechanical stress.

### 6.2. Necroptosis in Rheumatoid Arthritis

Rheumatoid arthritis (RA) is one of the most common chronic inflammatory diseases, which is characterized by joint inflammation and osteoclastogenesis. The key regulators of necroptosis, RIPK1, RIPK3 and MLKL were potently increased in the synovium of a collagen-induced arthritis mouse model [83], indicating that necroptosis might be involved in the pathogenesis of RA. RIPK1 inhibitor Nec-1 significantly decreased the expression of these key regulators and suppressed the expression of IL-17, IL-1β, IL-6 and TNFα in the mouse model [84]. Therefore, inhibiting RIPK1 might be a novel therapeutic approach for the treatment of RA.

### 6.3. Necroptosis in Kidney Injury

Necroptosis is also involved in kidney diseases. RIPK3 and MLKL are activated during acute kidney injury, where MLKL is expressed on the apical membrane of proximal tubules [85]. Thus, this suggests that the necrosome is an active mediator of kidney injury. Upon inhibition of kinase activities of RIPK1 and RIPK3 by specific inhibitors, membrane-located MLKL is reduced and cell viability is increased [85,86,87]. Furthermore, the SNx rat model for early progression of chronic kidney disease showed that RIPK1 inhibition by Nec-1 increased cell viability and reduced serum creatine and blood urea nitrogen [88]. It has also been demonstrated that crystal formation, as with calcium in kidney stones, activates the necrosome formation. Knockdown of RIPK3 or MLKL near completely inhibited the crystal-induced cytotoxicity [89]. Thus, inhibition of the necrosome has the potential to attenuate kidney injury of multiple sorts. Nonetheless, study of therapeutic efficacy of necroptosis inhibitors in preclinical models is necessary to further determine their usefulness in kidney diseases.

### 6.4. Necroptosis in Circulatory System Diseases

The necrosome has been found to be activated during cardiac injury. In both mouse and guinea-pig models studies of cardiac injury, it was found that inhibiting RIPK1 with Nec-1 was able to reduce infarct size and cardiac performance was preserved in these models, indicating that RIPK1 might be a feasible therapeutic target for cardiac injury [90,91]. Furthermore, it was also observed that under ER-stress conditions RIPK3 can migrate onto the ER and increase the production of reactive oxygen species (ROS) [92]. The elevated ROS lead to opening of the mitochondrial permeability transition pore, which causes cardiomyocytes to swell and rupture due to energy shortage. The level of ROS can be reduced to a normal level by RIPK3 depletion [92]. Based on the fact that RIPK3 induces ROS, targeting RIPK3 could be of therapeutic interest for prevention and treatment of cardiac injury.

The necrosome has also been observed to be activated in atherosclerotic plaques. Necrosome components, such as RIPK3 and MLKL, are highly expressed and activated in atherosclerotic plaques [93]. Furthermore, inhibiting RIPK1 with Nec-1 or downregulating RIPK3 were shown to be able to reduce advanced atherosclerotic lesions and the recruitment of macrophages [93,94], which indicates that targeting the different necrosome components could benefit patients with advanced atherosclerosis.

### 6.5. Necroptosis in Diabetes

Necroptosis was also found to be intimately involved in type 2 diabetes (T2D) with remarkable expression of the core components of necroptosis in specific obesity-associated tissues. High levels of RIPK1 and MLKL were detected in liver, adipose tissue and muscle in diabetic mice, whereas RIPK3 was only significantly upregulated in adipose tissue [95]. Furthermore, RIPK3 was also found to be highly expressed in the visceral white adipose tissue of obese humans [96]. T2D is a multifactorial metabolic disease characterized by the development of insulin resistance and glucose intolerance. Treating diabetic mice with RIPK1 inhibitor Nec-1 or depletion of MLKL prevented insulin resistance and glucose intolerance, but showed no effect on inflammation and cell death which are known as physiological consequences of necroptosis [95]. Both specific knockdown of RIPK1, RIPK3 or MLKL and the addition of chemical inhibitors Nec-1 or GSK’872 activated insulin-stimulated AKT signalling pathway, which plays an important role in glucose homeostasis. These findings suggest that the inhibitors of necrosome components might have therapeutic potentials for the treatment of T2D.

### 6.6. Necroptosis in Cancers

Necroptosis is often abrogated in multiple cancers. While the role of necroptosis in tumorigenesis has been well discussed [97], recent findings add to this picture. In colon cancer, the expression of RIPK1 and RIPK3 is significantly decreased due to hypoxia [98]. This suggests that down-regulation of necroptosis may contribute to tumorigenesis. Moreover, edelfosine, an antineoplastic drug, induced cell death in human glioblastoma cells by activating necroptosis. Inhibiting RIPK1 with Nec-1 or downregulating of RIPK3 abrogated edelfosine-induced cell death and increased cancer cell viability [99]. Also, it has been found that combining cell stress-inducing compounds with NF-κB inhibitors induced cell death by necroptosis in osteosarcoma cells [100]. This indicates that direct or indirect stimulation of necroptosis can be exploited for cancer treatment to induce cancer cell death. Furthermore, polymorphisms of RIPK1 have been found to be associated with differences in disease progression and correlated with overall survival and disease-free survival [101], suggesting that evaluation of RIPK1 polymorphism may be useful as a prognostic biomarker for tumor development.

Melanoma cell lines are resistant against necroptosis induction because they lack RIPK3 expression [102]. Overexpressing RIPK3 in these cell lines induced necroptosis via phosphorylation and activation of MLKL [102]. Furthermore, RIPK3-deficiency leads to activation of the NF-κB pathway in colon cancer. These cells have a high expression of IL-6, which is responsible for tumor progression through IL-6-induced Signal transducer and activator of transcription 3 signalling [103]. Thus, this shows a protective role for RIPK3 against tumor progression that could potentially be used for cancer therapy. Moreover, elevated expression of RIPK3 was found in cervical cancer cell lines [104]. PolyIC induced cervical cancer cells necroptosis and increased IL-1α secretion, which in turn activated dendritic cells. Subsequently, IL-1α could cause the release of IL-12 which is important for anti-tumor response [104]. This further illuminates the supportive role of RIPK3 in tumorigenesis. Although there is emerging evidence that necroptosis is abrogated in several cancer cells including melanoma, colon cancer, and cervical cancer, investigating the mechanisms behind this phenomenon would help find novel therapeutic strategies against these types of cancer.

## 7. Conclusions

It has been only 15 years since necroptosis was first described, but molecular insights into the mechanism and its relevance to diseases have been increased substantially during the last period. The necrosome components, MLKL, RIPK1 and RIPK3, are critical regulators of necroptotic cell death. MLKL functions as the executioner of necroptosis, depending on its phosphorylation, oligomerization and membrane translocation. RIPK1 acts rather as a traffic cop for different cell-death mechanisms but is not essential for inducing cell death. Current understandings illustrate a pathway in which RIPK3 activation, possibly mediated by RIPK1, induces MLKL activation leading to subsequent permeabilization of the plasma membrane and cell death. Despite the current knowledge of the activation and regulation of necroptosis, there remain many questions to be fully answered. For example, what are the specific events and enzymes involved in ubiquitination and activation of RIPK1/3, and what drives the exact effector mechanism of MLKL? What is the precise mechanism underlying the MLKL-mediated plasma membrane rupture during necroptosis? Does MLKL function as a carrier protein for some proteins, which do not contain the nuclear localization sequence and transport them from the cytoplasm into the nuclei to regulate necroptotic cell death? Due to the role of necroptosis in inducing inflammation and cell death, blocking necroptosis could be beneficial for pathologies including cell degeneration. On the other hand, activating necroptosis by regulating necrosome components might be a therapeutic strategy for cancer treatment, such as colon cancer, melanoma and cervical cancer. As the executioner of necroptosis, MLKL may be the most promising target for regulating necroptosis. While various small molecule inhibitors of RIPK1, RIPK3 and HSP90 are in under clinical development (Table 3), at present there are only two known MLKL inhibitors (NSA and GW806742X), and no MLKL inhibitor has been advanced into clinical trials, yet. In conclusion, current evidence suggests that agents that can specifically regulate MLKL activation may be of great interest in the treatment of necroptosis-related diseases.

## Figures and Tables

**Figure 1 cells-08-01486-f001:**
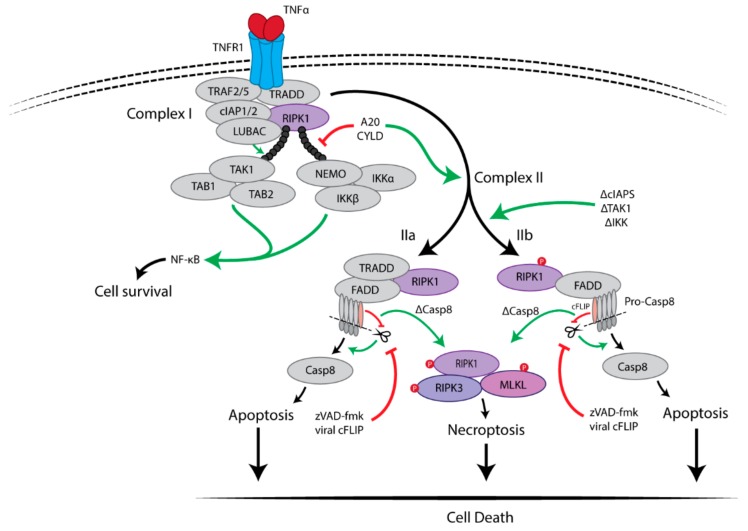
TNFR1-mediated survival and cell death pathways. The binding of TNFα to TNFR1 triggers multiple signaling pathways, including NF-κB, apoptosis and necroptosis. Upon stimulation, TNFα induces the formation of a membrane-associated complex I, which consists of RIPK1, TRADD, TRAF2/5, LUBAC and cIAP1/2. Within complex I, cIAP1/2 and LUBAC induce Lys63-linked polyubiquitination of RIPK1. The polyubiquitin chain of RIPK1 serves as a scaffold for further recruitment of IKK (IKKα, IKKβ and NEMO) and TAK1(TAK1, TAB1 and TAB2) complexes, eventually leading to activation of NF-κB pathway and cell survival. Deubiquitination of RIPK1 by CYLD or A20 induces the dissociation of TRADD and RIPK1 from TNFR1, which leads to the formation of either of complex IIa or complex IIb. FADD and pro-caspase-8 are recruited to TRADD and RIPK1 to form complex IIa, resulting in the activation of caspase-8 by oligomerization and cleavage. In the absence of cIAP1/2, TAK1 or IKK complex, complex IIb, which contains RIPK1, FADD and pro-caspase-8 except TRADD, is formed and then activate caspase-8. Then, caspase-8 induces apoptosis. RIPK3-dependent necroptosis is induced when caspase-8 activity is blocked, for example by cFLIP or the pan-caspase inhibitor zVAD-fmk.

**Figure 2 cells-08-01486-f002:**
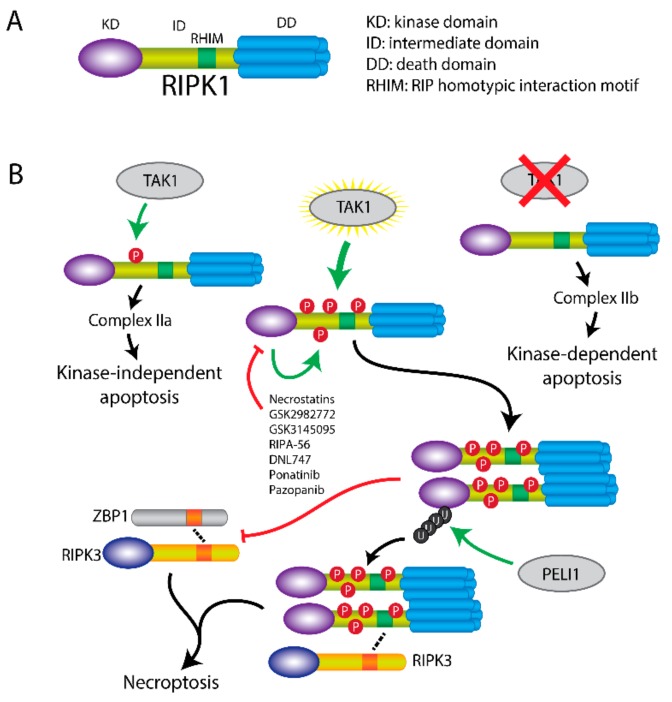
Cell death induction by RIPK1 regulation. (**A**) Scheme of RIPK1 structure. RIPK1 consists of an N-terminal kinase domain (KD), a RHIM containing intermediate domain (ID), and a C-terminal death domain (DD). (**B**) RIPK1-mediated cell death is regulated by TAK1. TAK1-induced phosphorylation of ID domain within RIPK1 initiates RIPK1-independent apoptosis via complex IIa. In the absence of TAK1, RIPK1-dependent apoptosis is induced via complex IIb. When RIPK1 is hyperphosphorylated by TAK1, RIPK3-dependent necroptosis is induced through DD dimerization. Necroptosis can be inhibited by RIPK1 inhibitors, such as necrostatins, GSK2982772, GSK3145095, RIPA-56, DNL747, ponatinib and pazopanib. E3-ligase PELI1 induces polyubiquitination of RIPK1, which in turn regulates the interaction of RIPK1 with RIPK3 and promotes necroptosis. Besides induction of cell death, RIPK1 also exerts necroptosis-inhibitory functions by inhibiting RIPK3 activation through other RHIM containing proteins, such as ZBP1.

**Figure 3 cells-08-01486-f003:**
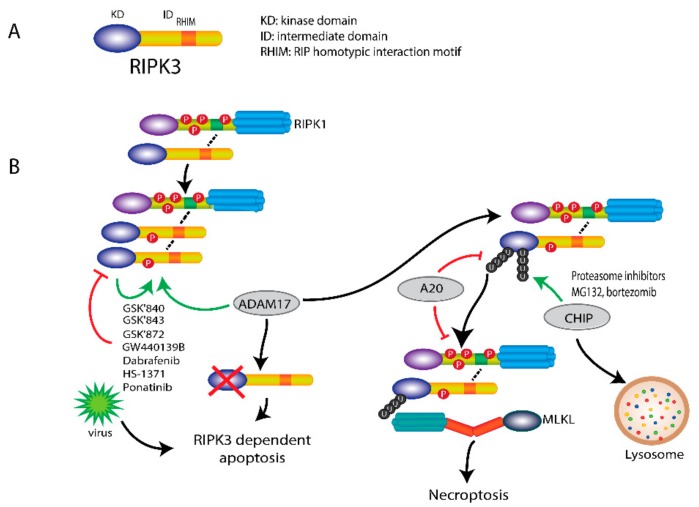
RIPK3-mediated necroptosis. (**A**) Scheme of RIPK3 structure. RIPK3 consists of an N-terminal kinase domain (KD), a RHIM containing intermediate domain (ID). (**B**) The association of RIPK1 with RIPK3 induces the phosphorylation and activation of RIPK3 by its kinase-activity and ADAM17. Upon inhibition of RIPK3 by specific inhibitors, such as GSK’840, GSK’843, GSK’872, GW440139B, HS-1371, ponatinib and dabrafenib (a B-raf inhibitor), RIPK3-dependent apoptosis is initiated, which can also be induced by viruses. Ubiquitination of RIPK3 also regulates RIPK3 activity. Upon necroptosis induction, RIPK3 is ubiquitinated at Lys5. Deubiquitination of RIPK3 Lys5 by A20 can inhibit necroptosis by blocking the interaction of RIPK1 and RIPK3 and the necrosome formation. In addition, ubiquitination of RIPK3 by CHIP at Lys55 and Lys363 can also block necroptosis by triggering lysosomal degradation. Proteasome inhibitors MG132 and bortezomib can induce necroptosis through the accumulation of polyubiquitination of RIPK3 at Lys264 without caspase-8 inhibition.

**Figure 4 cells-08-01486-f004:**
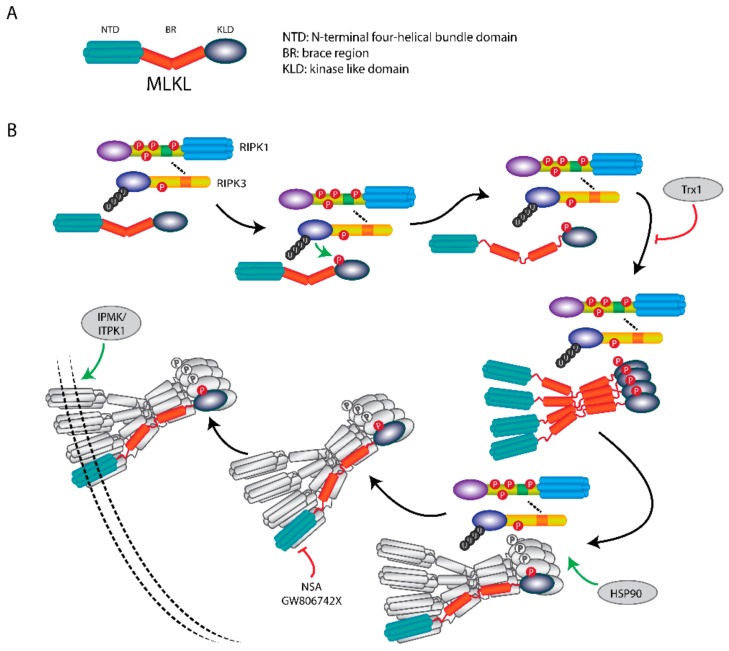
MLKL translocation and membrane association induce necroptosis. (**A**) Scheme of MLKL structure. MLKL contains an N-terminal α-helices domain (NTD), a brace region (BR), and a C-terminals pseudokinase domain (PKD). (**B**) Upon necroptosis induction, MLKL forms a necrosome with RIPK1 and RIPK3, where it is activated by RIPK3-mediated phosphorylation. The phosphorylation of MLKL leads to release of the BR and NTD, followed by the formation of tetramers and subsequent octamers. After the MLKL octamer is dissociated from the necrosome, it translocates to the plasma membrane to disrupt membrane integrity and induces necroptosis. The membrane association of phosphorylated-MLKL can be regulated by IP kinases IPMK and ITPK1. HSP90 is indispensable for necrosome formation. Necroptosis can be inhibited by MLKL inhibitors NSA, GW806742X and protein Trx1.

**Table 1 cells-08-01486-t001:** Comparison between cellular characteristics in apoptosis, necroptosis and necrosis.

Apoptosis	Necroptosis	Necrosis
Regulated	Regulated	Unregulated
Triggered by specific cytokines	Triggered by specific cytokines	Induced by damaging environmental stress (such as extreme physiological stress and viral or toxin-mediated infections)
Shrinkage of cell	Rapid swelling of cell	Cell swelling
-	Swelling of organelles	Swelling of organelles
Blebbing of plasma membrane	Disruption of plasma membrane	Loss of membrane integrity
Formation of apoptotic bodies	Organelle breakdown	
-	Release of cellular contents	Release of cellular contents
-	Pro-inflammatory response	Pro-inflammatory response

**Table 2 cells-08-01486-t002:** Necroptosis inhibitors.

Compound Name	Target	References
Necrostatins (necrostatin-1, 7-Cl-O-necrostatin)	RIPK1	[1,41]
GSK2982772	RIPK1	[42]
GSK3145095	RIPK1	(Clinicaltrials.gov)
RIPA-56	RIPK1	[40]
Ponatinib	RIPK1/RIPK3	[43]
Pazopanib	RIPK1	[43]
DNL747	RIPK1	(Clinicaltrials.gov)
GSK’840, GSK’843 and GSK’872	RIPK3	[44,45]
Dabrafenib	RIPK3	[46]
HS-1371	RIPK3	[47]
GW806742X	MLKL	[48]
Necrosulfonamide (NSA)	MLKL	[49]
17AAG	HSP90	[50]

**Table 3 cells-08-01486-t003:** Necroptosis inhibitors under clinical development.

Compound Name	Target	Phase	Disease Condition	Identifier	Study Type	Status (Oct 2019)
GSK2982772	RIPK1	II	Ulcerative colitis	NCT02903966	With placebo	Completed
		I	Rheumatoid arthritis	NCT02858492	With placebo	Completed
		II	Psoriasis	NCT02776033	With placebo	Completed
GSK3145095		I	Solid tumors	NCT03681951	With Pembrolizumab	Terminated
Dabrafenib	RIPK3	IV	Melanoma Non-small cell lung cancer Solid tumor Rare cancers High grade glioma	NCT03340506	With trametinib	Recruiting
		II	Metastatic colorectal cancer	NCT03668431	With trametinib and PDR001	Recruiting
		II	Melanoma	NCT01682213	Single agent	Completed
		II	BRAF Mutation-Positive Malignant Melanoma and Brain Metastases	NCT02974803	With trametinib and stereotactic radiation	Active, not recruiting
		II	Metastatic Melanoma (Carrying BRAF V600 Mutation)	NCT02052193	With vemurafenib	Terminated
		III	Malignant Melanoma	NCT03551626	With trametinib	Recruiting
Ponatinib	RIPK/RIPK3		Chronic Myeloid Leukemia (CML) Philadelphia Chromosome Positive Acute Lymphoblastic Leukemia (Ph+ ALL)	NCT01592136	Expanded Access	Approved for marketing
		II	Non-small cell lung cancer	NCT01813734	Single agent	Completed
		II	Leukemia	NCT01570868	Single agent	Terminated
		II/III	Non-small cell lung cancer head and neck Cancer	NCT01761747	Single agent	Terminated
		I/II	Acute myeloid leukemia	NCT02428543	With cytarabine	Recruiting
		II	Medullary thyroid cancer	NCT03838692	Single agent	Not yet recruiting
Pazopanib	RIPK1	I	Renal cell carcinoma Soft tissue sarcoma Metastatic disease	NCT02795819	With AR-42	Terminated
		II	Renal Cell carcinoma	NCT01545817	Pazopanib followed by everolimus	Terminated
DNL747	RIPK1	I	AD	NCT03757325	With Placebo	Recruiting
		I	ALS	NCT03757351	With Placebo	Recruiting
17AAG	HSP90	II	Anaplastic Large Cell Lymphoma Recurrent Adult Hodgkin Lymphoma Recurrent Mantle Cell Lymphoma	NCT00117988	Single agent	Completed
		I	Unspecified Adult Solid Tumor, Protocol Specific	NCT00121264	With sorafenib tosylate	Completed
		I	Relapsed or Refractory Hematologic Cancer	NCT00103272	With bortezomib	Terminated
IPI-504	HSP90	I	Solid Tumors	NCT00606814	With Docetaxel	Completed
		I	Multiple Myeloma	NCT00113204	Single agent	Completed
		II	Prostate Cancer	NCT00564928	Single agent	Completed

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
