# Peer review of "Molecular Insights into the Mechanism of Necroptosis: The Necrosome as a Potential Therapeutic Target"

_cells, 2019, doi:10.3390/cells8121486_

Round 1

Reviewer 1 Report

Following please find our suggestions to your cells review paper entitiled, “Molecular Insights into the Mechanism of Necroptosis: the necrosome as a potential therapeutic target” by Frank Redegeld group including leading authors, Jing Chen, Renate Kos, and Johan Garssen. Necroptosis is an important programmed cell death that is essential for animal development, cell defense to a variety stimulates, and immune responses to cancer cells and degenerative cells. While apoptosis is so important to cell function and number control in animal development, necroptosis seems relatively poorly understood. Jing Chen’s review manuscript provides a better understanding of current views of how RIPK1, RIPK3 and MLDL can function as a molecular mechanism to transduce cell death program via necroptosis, and its potential implication in pharmacological intervention for anti-cancer or other necroptosis-associated human diseases. Frank Redegeld group has been studied the important problem, but we would like to also raise two concerns about this review manuscript. Firstly, it seems that RIPK 1 dependent pro-survival signaling complex I could promote cell survival via TAK activated NF-kB pathway; whereas, RIPK 1 dependent cell death-inducing signaling complex II via RIPK 3 and MLKL. However, the authors also list a table of current necroptosis inhibitors, such as necrostatins and Ponatinib. It seems reasonable for readers by connecting the molecular pharmacological points to increase clarity, such as point to where and how these necroptosis inhibitors work by conjugation with their molecular pathway, i.e. label them in main text and figures. Secondly, it seems that the role of necroptosis in cancers and major neurodegenerative diseases has given the high interest in treating cellular responses to diseases. However, how necroptosis participates in the cellular vulnerability within a variety of physiological or different neurodegenerative diseases and cancers disease conditions that is not reviewed. Authors are encouraged to refer these readers who might be interests in more detailed of RIPK family or necroptosis in specific cancer or specific neurodegenerative diseases to recent reviews such as Najafov Ayaz group. “Necroptosis and Cancer, 2017” and Sang, Tzu-Kang group “Neuronal cell death mechanisms in major neurodegenerative disease”. Overall, we are delighted to review Jing's review given the current understanding of necroptosis.

Author Response

We thank the reviewer for the positive evaluation of the manuscript and the suggestions to further improve it. 

In answer to the specific comments:

It seems reasonable for readers by connecting the molecular pharmacological points to increase clarity, such as point to where and how these necroptosis inhibitors work by conjugation with their molecular pathway, i.e. label them in main text and figures. Answer: We have now added the inhibitors into the figures to show where interaction will take place. Authors are encouraged to refer these readers who might be interests in more detailed of RIPK family or necroptosis in specific cancer or specific neurodegenerative diseases to recent reviews such as Najafov Ayaz group. “Necroptosis and Cancer, 2017” and Sang, Tzu-Kang group “Neuronal cell death mechanisms in major neurodegenerative disease”. Answer: we have added these references to the text and reference list.

Reviewer 2 Report

In this review paper, the authors discuss an interesting and timely topic, namely “Molecular insights into the mechanism of necroptosis”. The review begins with an overall description of necroptosis including a death-inducing signaling complex formation. Thereafter, the roles of necroptosis regulators, RIPK1, RIPK3, MLKL, and the information of their inhibitors are described. In the last part, the authors discuss about the association of necroptosis with various diseases.

Overall, the manuscript well written and provide a comprehensive overview of the related literature on the topic.

Miner Points

Line 61: Please add reference

Line 81: and the initial signal transduction.[11]. (please remove “the period” in front of [11])

Line 111, 150: ZVAD-fmk  change to  zVAD-fmk

Line 145: cFLIPL, cFLIPS change to cFLIPL , cFLIPs

Line 194: “Several studies have been attempted to…” ( add related references)

Line 277: tfag?

Line 280-282: “MLKL is essential for necroptosis and……” Not clear, please rephrase

Line 362: “…complemented cells the membrane disruption” Please add comma between cells and the membrane

Author Response

We thank the reviewer for the positive evaluation of the manuscript and the suggestions to further improve it. 

Line 61: Please add reference

Done

Line 81: and the initial signal transduction.[11]. (please remove “the period” in front of [11])

Done

Line 111, 150: ZVAD-fmk  change to  zVAD-fmk

Done

Line 145: cFLIPL, cFLIPS change to cFLIPL , cFLIPs

Done

Line 194: “Several studies have been attempted to…” ( add related references)

Done

Line 277: tfag?

Corrected in tag

Line 280-282: “MLKL is essential for necroptosis and……” Not clear, please rephrase

Rephrased into: After activation of RIPK3, in the absence of caspase-8 activity, MLKL is recruited to RIPK3 and induces necroptosis

Line 362: “…complemented cells the membrane disruption” Please add comma between cells and the membrane

Done